# Classifier uncertainty: evidence, potential impact, and probabilistic treatment

Niklas Tötsch and Daniel Hoffmann

Faculty of Biology, University of Duisburg-Essen, Essen, Germany

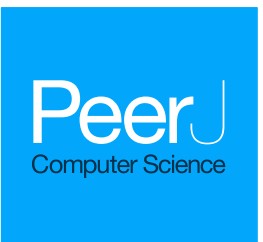

## ABSTRACT

Classifiers are often tested on relatively small data sets, which should lead to uncertain performance metrics. Nevertheless, these metrics are usually taken at face value. We present an approach to quantify the uncertainty of classification performance metrics, based on a probability model of the confusion matrix. Application of our approach to classifiers from the scientific literature and a classification competition shows that uncertainties can be surprisingly large and limit performance evaluation. In fact, some published classifiers may be misleading. The application of our approach is simple and requires only the confusion matrix. It is agnostic of the underlying classifier. Our method can also be used for the estimation of sample sizes that achieve a desired precision of a performance metric.

## INTRODUCTION

Classifiers are ubiquitous in science and every aspect of life. They can be based on experiments, simulations, mathematical models or even expert judgement. The recent rise of machine learning has further increased their importance. But machine learning practitioners are by far not the only ones who should be concerned by the quality of classifiers. Classifiers are often used to make decisions with far-reaching consequences. In medicine, a therapy might be chosen based on a prediction of treatment outcome. In court, a defendant might be considered guilty or not based on forensic tests. Therefore, it is crucial to assess how well classifiers work.

In a binary classification task, results are presented in a $2 \times 2$ confusion matrix (CM), comprising the numbers of true positive (TP), false negative (FN), true negative (TN) and false positive (FP) predictions.

$$CM = \begin{bmatrix} TP & FN \\ FP & TN \end{bmatrix} \tag{1}$$

The confusion matrix contains all necessary information to determine metrics which are used to evaluate the performance of a classifier. Popular examples are accuracy (ACC), true positive rate (TPR), and true negative rate (TNR).

Corresponding author
Niklas Tötsch,
niklas.toetsch@uni-due.de

$$\text{ACC} = \frac{\text{TP} + \text{TN}}{\text{TP} + \text{FN} + \text{FP} + \text{TN}} \tag{2}$$

$$\text{TPR} = \frac{\text{TP}}{\text{TP} + \text{FN}} \tag{3}$$

$$\text{TNR} = \frac{\text{TN}}{\text{TN} + \text{FP}} \tag{4}$$

These are given as precise numbers, irrespective of the sample sizes (*Ns*) used for their calculation in performance tests. This is problematic especially in fields such as biology or medicine, where data collection is often expensive, tedious, or limited by ethical concerns, leading often to small *Ns*. In this study we demonstrate that in those cases the uncertainty of the CM entries cannot be neglected, which in turn makes all performance metrics derived from the CM uncertain, too. In the light of the ongoing replication crisis (*Baker, 2016*), it is plausible that negligence of the metric uncertainty impedes reproducible classification experiments.

There is a lack of awareness of this problem, especially outside the machine learning community. One often encounters discussions of classifier performance lacking any statistical analysis of the validity in the literature. If there is a statistical analysis it usually relies on frequentist methods such as confidence intervals for the metrics or null hypothesis significance testing (NHST) to determine if a classifier is truly better than random guessing. NHST "must be viewed as approximate, heuristic tests, rather than as rigorously correct statistical methods" (*Dietterich, 1998*).

Bayesian methods can be valuable alternatives (*Benavoli et al., 2017*). To properly account for the uncertainty, we have to replace the point estimates in the CM and all dependent performance metrics by probability distributions. Correct and incorrect classifications are outcomes of a Binomial experiment (*Brodersen et al., 2010a*). Therefore, Brodersen et al. model ACC with a beta-binomial distribution (BBD)

$$\text{ACC} \sim \text{Beta}(\text{TP} + \text{TN} + 1, \text{FP} + \text{FN} + 1). \tag{5}$$

Some of the more complex metrics, such as balanced accuracy, can be described by combining two BBDs (*Brodersen et al., 2010a*).

Caelen presented a Bayesian interpretation of the CM (*Caelen, 2017*). This elegant approach, based on a single Dirichlet-multinomial distribution, allows to replace the count data of the confusion matrix with distributions which account for the uncertainty.

$$\text{CM} \sim \text{Mult}(\theta, N) \tag{6}$$

$$\theta \sim \text{Dirichlet}((1, 1, 1, 1)) \tag{7}$$

where $\theta = [\theta_{\text{TP}}, \theta_{\text{FN}}, \theta_{\text{TN}}, \theta_{\text{FP}}]$ is the confusion probability matrix which represents the probabilities to draw each entry of the CM. The major advantage of Caelen's approach over the one presented by Brodersen lies in a complete description of the CM. From there, all metrics can be computed directly, even those that cannot simply be described as BBD.

Caelen calculates metric distributions from confusion matrices that are sampled according to Eq. (6). Here, we demonstrate that this approach is flawed and derive a correct model. Whereas previous studies focused on the statistical methods, we prove that classifier performance in many peer-reviewed publications is highly uncertain. We studied a variety of classifiers from the chemical, biological and medicinal literature and found cases where it is not clear if the classifier is better than random guessing. Additionally, we investigate metric uncertainty in a Kaggle machine learning competition where sample size is relatively large but a precise estimate of the metrics is required. In order to help non-statisticians to deal with these problems in the future, we derive a rule for sample size determination and offer a free, simple to use webtool to determine metric uncertainty.

## METHODS

### Model

The confusion probability matrix ($\theta$), that is the probabilities to generate entries of a confusion matrix, can be derived if prevalence ($\phi$), TPR and TNR are known (*Kruschke, 2015a*).

$$\theta_{TP} = TPR \cdot \phi \tag{8}$$

$$\theta_{FN} = (1 - TPR) \cdot \phi \tag{9}$$

$$\theta_{TN} = TNR \cdot (1 - \phi) \tag{10}$$

$$\theta_{FP} = (1 - TNR) \cdot (1 - \phi) \tag{11}$$

The idea that these metrics can also be inferred from data, propagating the uncertainty, is the starting point of the present study. Using three BBDs, one for each of $\phi$, TPR and TNR, we can express all entries of the CM (Fig. 1). Since $\phi$, TPR and TNR are distributions, the entries of cpm $[\theta_{TP}, \theta_{FN}, \theta_{TN}, \theta_{FP}]$ are too. Based on $\theta$ we calculate all other metrics of interest.

For the following Bayesian treatment we use the Laplace prior, Beta($\alpha = 1$, $\beta = 1$), for $\phi$, TPR and TNR because its uniform distribution introduces no bias, which makes it suitable for any classification problem. It is noteworthy that a flat prior on $\phi$, TPR and TNR leads to non-flat priors on other metrics (Section S1). We discuss two additional objective priors in the Supplemental Material. If additional knowledge is available, based for example, on the experimental setup of the classifier, it should be incorporated in the prior. Here, we refrain from using informative priors to keep the method generally applicable.

Our approach is quite similar to Caelen's but has distinct advantages. First, $\phi$, TPR and TNR are common metrics; thus prior selection is easier. Second, our model clearly distinguishes data intrinsic $\phi$ from the classifier intrinsic measures TPR and TNR. Consequently, our approach allows to "exchange" $\phi$. This is useful if the prevalence of the

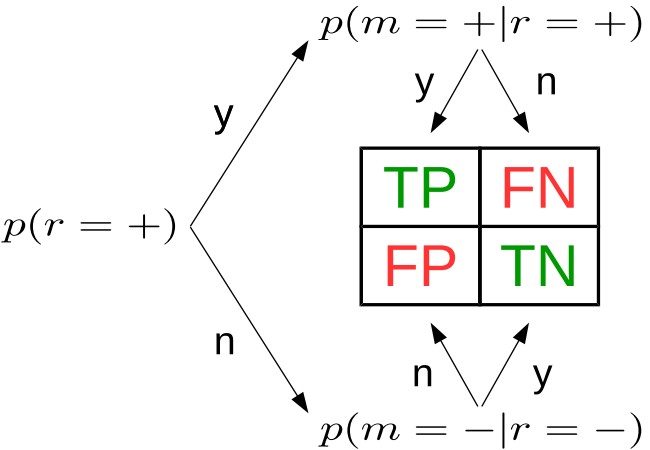

**Figure 1 Three beta-binomial distributions p(·)—prevalence (left), true positive rate (top), true negative rate (bottom)—define the confusion matrix.** Based on them, all entries of the CM can be expressed as distributions with explicit uncertainty due to limited sample size. In the figure, $r$ stands for reference, $m$ for model.

test set differs from the prevalence of the population the classifier will be applied to in production. Such a scenario is common in medical tests where $\phi$ is very low in the general population. To increase the sample size of positive cases in the test set without inflating the number of negative ones, $\phi$ differs from the general population. Using a Dirichlet-multinomial distribution, it is not straightforward to evaluate a classifier for a different $\phi$. If the data set was designed to contain a specified fraction of positive and negative instances, $\phi$ is known exactly (Section S2). This scenario is easy to implement in our model but not in Caelen's.

Depending on the context, $\phi$ may have two meanings. If one is interested in a population, $\phi$ describes how common fulfillment of the positive criterion is. For an individual, for example, a patient, $\phi$ can be considered the prior. If additional information was available for this subject, such as results of previous tests, $\phi$ would differ from the prevalence in the general population. This prior can be updated with TPR and TNR, representing the likelihood, to yield the posterior for the individual.

## Measuring true rather than empirical uncertainty

Measuring true rather than empirical uncertainty Bayesian models allow posterior predictions. In our case, posterior predictions would be synthetic confusion matrices $V$, which can be generated from a multinomial distribution (Eq. (6)).

This approach is equivalent to a combination of two/three binomial distributions as shown in Fig. 1 but slightly more elegant for posterior predictions. Caelen samples many $V$ to obtain metric distributions, which requires a choice of sample size $N$. Caelen uses the $N$ of the original CM the parameters have been inferred from. This is not satisfying because in this way only the empirical distribution of the metrics for a given $N$ is generated, not the true distribution of the metrics. Consider the example of CM = (TP, TN, FP, FN) = (1, 0, 0, 0), that is, $N = 1$. We will consider this classifier's ACC.

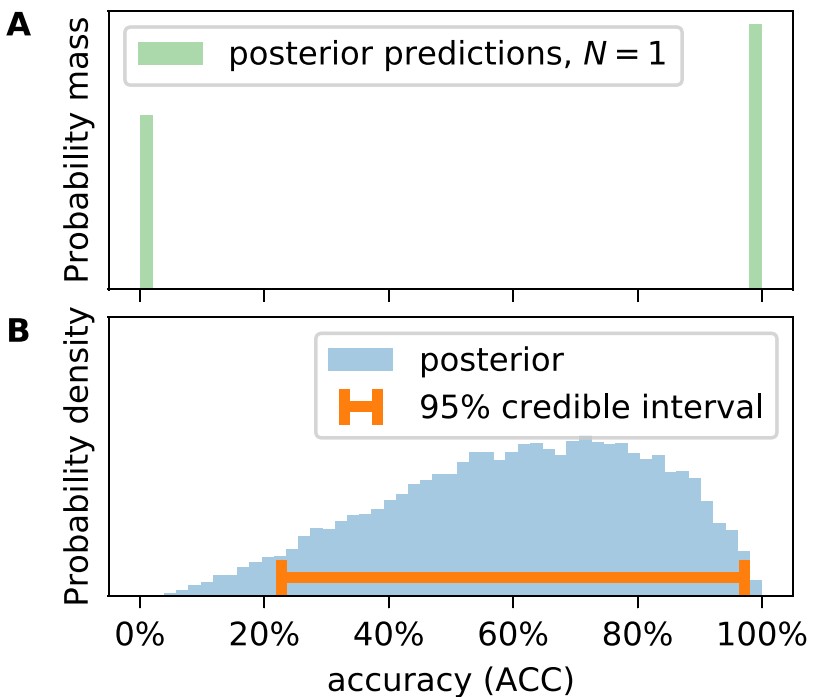

**Figure 2 Calculating accuracy (ACC) on posterior predictions of the confusion matrix yields a discrete distribution (A), representing expected observations of the metric at given sample size (_N_). Posterior distributions (B) of the metric must be calculated from the inferred entries of the confusion probability matrix (θ) as outlined in the text.**

Caelen's approach leads to a discrete distribution of the accuracy allowing only 0 and 1 (Fig. 2A). There was one correct prediction in the original CM, therefore it is impossible that the accuracy is 0. In other words, the probability mass at ACC = 0 should be strictly 0. If one is interested in the true continuous posterior distribution of a metric, one must calculate it from θ directly (Fig. 2B). We prove in Section S4 that Caelen's approach systematically overestimates the variance in metric distributions.

We still consider Caelen's way of calculating metrics extremely useful since it allows to tackle the problem of reproducibility. Generating synthetic _V_ according to Eq. (6) allows us to estimate what would happen if multiple researchers applied the same classifier to different data sets of size _N_ and reported the corresponding CMs and metrics. Figure 2 shows that they might report completely different values of a metric if _N_ is small. Under these circumstances, classification experiments are not reproducible.

## Metric uncertainty equals credible interval length

If there is little data available, posterior distributions are broad. We define metric uncertainty (MU) as the length of the 95% highest posterior density interval ("credible interval"). There is a 95% likelihood that the metric is within this credible interval (bottom of Fig. 2B). In Section S5, we prove that the uncertainty of φ, TPR, TNR, and other metrics is dependent on $\frac{1}{\sqrt{N}}$.

## Implementation

Since the beta distribution is the conjugate prior of the binomial distribution, the posterior distribution can be derived analytically. There is no need for Markov chain Monte Carlo sampling. This is merely a convenience, our approach would work with any prior. To calculate metrics, we sampled 20,000 data points. Splitting these data points into two arrays of equal length, we use PyMC's implementation of the Gelman–Rubin diagnostics ($R_c < 1.01$) to verify that the posterior distribution is properly sampled (*Gelman & Rubin, 1992*; *Brooks & Gelman, 1998*; *Salvatier, Wiecki & Fonnesbeck, 2016*).

The implementation of our model in Python can be found at GitHub (https://github.com/niklastoe/classifier_metric_uncertainty).

# RESULTS AND DISCUSSION

## Classifier examples from the literature

To assess the uncertainty in classifier performance in the scientific literature, we searched Google Images for binary confusion matrices from peer reviewed publications in the area of chemistry, biology and medicine with less than 500 samples in the test set. We collected 24 classifiers; confusion matrices and the references to the publications are listed in Table S1. Publications are indexed with numbers. If more than one classifier is presented in one publication, a character is added. Some of these classifiers are based on statistical models of available data. Others are based on simulations. The majority of publications describe the development of a new experimental approach followed by a statistical model that transforms the experimental outcome into a classification. Classifiers come from diverse fields, for example, chemical detection (adulterants in palm oil or cocaine, mycotoxins in cereals) or prediction of inhibitors of amyloid-aggregation or enzymes. The smallest sample size was 8, the largest 350.

While the resources invested in the development of these classifiers must have been considerable, their performance had not been thoroughly evaluated. Specifically, only for a single classifier the uncertainty had been quantified by calculating confidence intervals. In some of the literature examples, we also noted severe problems unrelated to small *N*. Due to usage of ACC for imbalanced data sets and mixing of train and test data sets for reported metrics, the performance of some classifiers was overrated. These problems have been addressed previously (*Chicco, 2017*). In this study, we evaluate classifiers on metrics which are invariant to class imbalance and rely exclusively on test data sets.

Our selection may not in all aspects be representative of published classifiers in any field. However, the negligence of metric uncertainty observed in this selection is not exceptional. Our choice of biology, chemistry, and medicine as scientific domain was based on our relative familiarity with those fields. While in this area small sample sizes are common (due to costly data collection), this problem is probably not limited to this domain.

## Metrics are broadly distributed

Typically, classifier metrics are reported as single numerical values (often to one or more decimals) without indication of uncertainty. However, the true MUs of classifiers in our collection are too large to be ignored (Fig. 3B). Often, MU is greater than 20 percentage

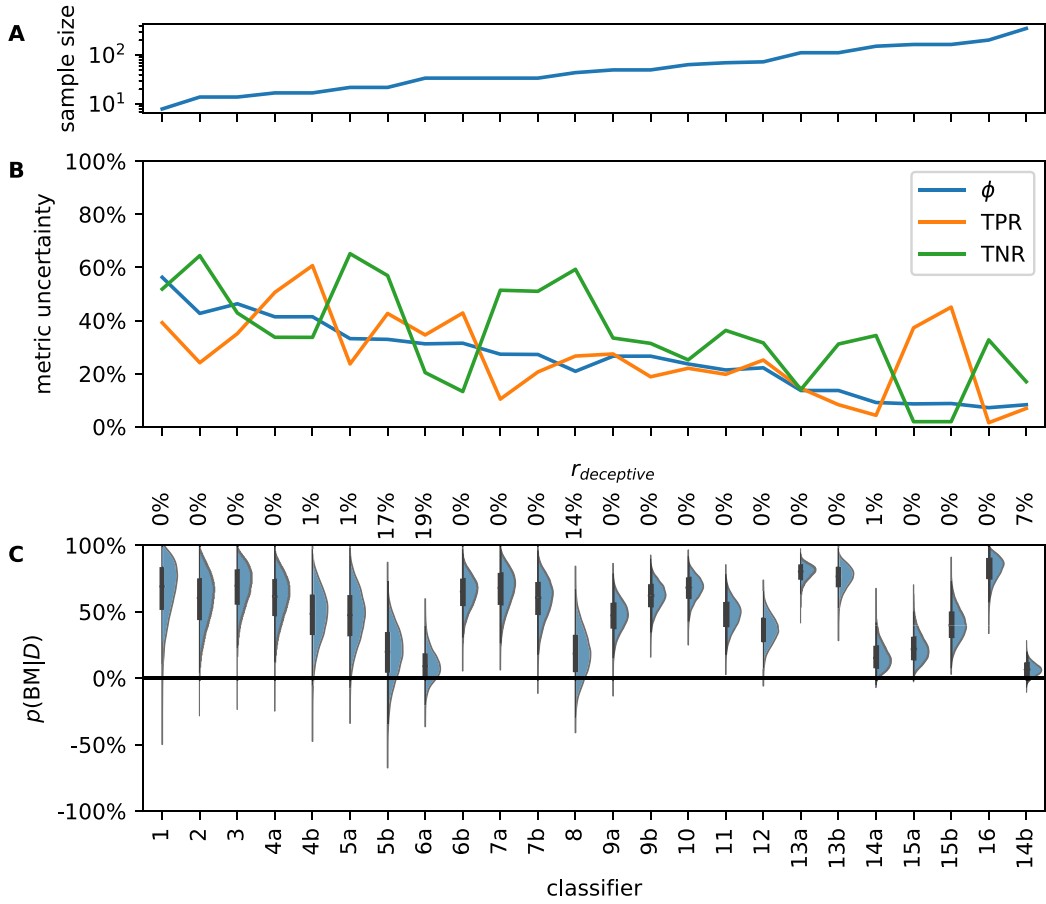

**Figure 3** **Analysis of literature examples. Classifiers are sorted by ascending sample size which ranges from 8 to 350 (A). Metric uncertainty (MU) for prevalence (ϕ), true positive rate (TPR), and true negative rate (TNR) is large and decreases with sample size (B). Since MU is determined by the length of the 95% highest posterior density interval, the theoretical upper limit is 95% (in which case nothing is known about the metric). If MU was 0%, the corresponding metric would be known at infinite precision. Posterior distributions of bookmaker informedness (BM) are broad due to small test sets in the literature examples (C).** Some classifiers have considerable posterior density in the negative region; these classifiers could be misinformative. Percentages along top margin are $r_{deceptive}$ values Eq. (14), the probability that a classifier is worse than random guessing.

points, sometimes exceeding 60 percentage points. In general, MU in all three observed metrics declines as *N* increases. The decrease is not monotonous because MU also depends on the value of the metric (Section S5).

The MUs we show in Fig. 3B were obtained from θ. As mentioned above, metrics calculated from empirically observed confusion matrices of the same classifier would vary even more. Thus, if an independent lab tried to reproduce CM for, say, example 7a, with a much larger sample size, TNR values of 90% or 50% would not be surprising, although the value given in the article is 75%.

It is possible that we underrate some classifiers. If a metric should have a more informative prior than the Laplace prior we used, for example, due to previous experience or convincing theoretical foundations, the posterior could also be more narrowly defined.

**Table 1 Confusion matrix of the cocaine purity classifier 7a. *r* stands for reference, *m* for model.**

|  | *r* = high | *r* = low |
|---|---|---|
| *m* = high | 26 | 2 |
| *m* = low | 0 | 6 |

### Metric uncertainty limits confidence in high-stakes application of classifiers

In the following, we discuss in greater detail MU for one classifier where the consequences of misclassification are dramatic and understandable to non-experts. Classifier 7a is a new method to predict cocaine purity based on a "simple, rapid and non-destructive" experiment followed by mathematical analysis. The authors stress the importance of such a method for forensic experts and criminal investigators. Predictions are compared to a destructive and more elaborate experimental reference. Prosecutors in countries such as Spain may consider purity as evidence of the intent to traffic a drug, presumably resulting in more severe punishments (http://www.emcdda.europa.eu/system/files/publications/3573/Trafficking-penalties.pdf, accessed 3 December 2019 1:55 pm CEST). Consequently, a FP would result in a wrongful charge or conviction causing severe stress and eventually imprisonment for the accused. A FN on the other hand might lead to an inadequately mild sentence. Moreover, one could also consider the scenario of drug checking. In some cities, such as Zurich, Switzerland, social services offer to analyze drugs to prevent harm from substance abuse due to unexpectedly high purity or toxic cutting agents (https://www.saferparty.ch/worum-gehts.html, accessed 9 June 2020 at 3:42 pm CEST). In this context, a FN could lead to an overdose due to the underestimated purity.

The confusion matrix in Table 1 is transcribed from the original publication. We do not know whether their method was used for drug checking or in court (at least the authors received the samples from the local police department). If it was, could it be trusted by a forensic expert, judge, or member of the jury? The posterior distribution of the TPR (Fig. 4A) answers this question probabilistically. The point estimate from CM would be TPR = 100% but due to small $N$, the uncertainty is large. The credible interval spans from 89% to almost 100% although not a single FN has been observed in the test set.

Now consider TNR (Fig. 4B). Since there are only eight low purity cocaine samples, the uncertainty is much larger. While the point estimate would be TNR = 75%, the credible interval is 43–95%. It is possible, although unlikely, that the classifier would generate more FP than TN. This would translate into more wrongful convictions than correct acquittals for possessing cocaine with high purity if this method was used as main evidence in court.

Our approach would hopefully lead to more cautious use of little tested classifiers. Imagine two scenarios. In the first, a judge is told that the forensic method has a TPR of 100% and a TNR of 75%. In the second, she is told that it has an estimated TPR of 89–100% and an estimated TNR of 43–95%. In the latter, the judge would be more hesitant to base her verdict on the classifier.

We do not know if $\phi$ in the test set is representative of the prevalence of drug samples in criminal cases. Therefore, we cannot reasonably estimate the distribution of

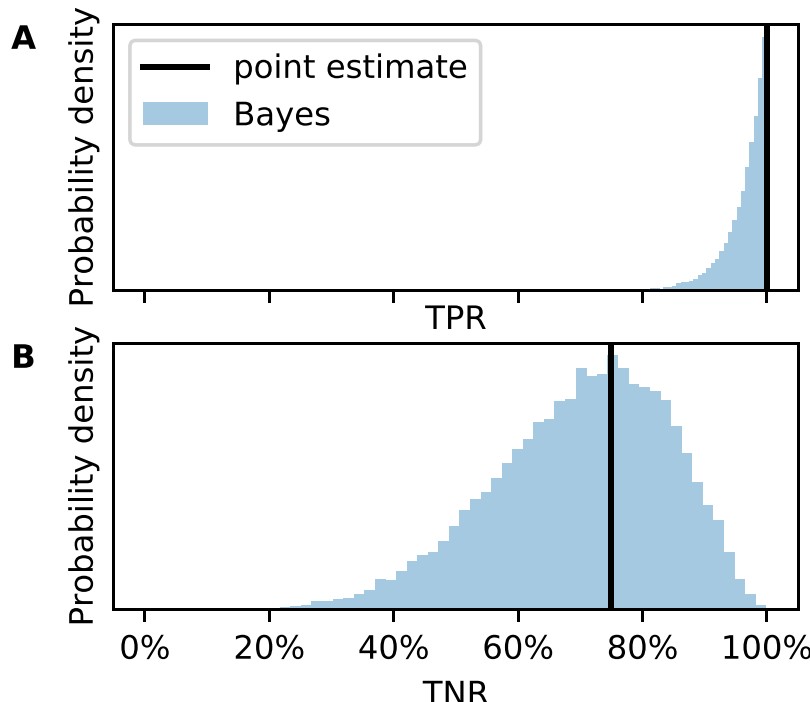

**Figure 4 Metric uncertainty for cocaine purity classifier 7a.** Posterior distributions for (A) true positive rate (TPR) and (B) true negative rate (TNR).

probabilities of wrongfully harsh/lax sentences. For a meaningful assessment of evidence, both ϕ and MU should be taken into account. Our approach facilitates such an analysis.

### Some published classifiers might be deceptive

As classification problems vary greatly so does the relevance of different metrics, depending on whether FN or FP are more or less acceptable. Often, classifier development requires a tradeoff between FN or FP. In this respect, bookmaker informedness (BM) is of interest because it combines both in a single metric without weighting and measures the probability of an informed prediction *Powers (2011)*.

$$BM = TPR + TNR - 100\%$$ (12)

If BM = 100%, prediction is perfect and the classifier is fully informed. BM = 0% means that the classifier is no better than random guessing and BM = −100% shows total disagreement, that is, the predictor is wrong every single time. Figure 3C shows the posterior distributions of BM for the collected examples from literature. Due to small *N*, they are broad. Therefore, it is uncertain how much better the classifiers are compared to random guessing. Several classifiers have considerable probability density in the negative region, that is, it is possible that they are weakly deceptive.

We define the probabilities that a given classifier is informative or deceptive

$$r_{\text{informative}} = \int_{0\%}^{100\%} p(\text{BM}|D)d\theta$$ (13)

$$r_{\text{deceptive}} = \int_{-100\%}^{0\%} p(\text{BM}|D)d\theta. \tag{14}$$

We determined $r_{\text{deceptive}}$ for all literature examples (Fig. 3C, top). Four classifiers have a considerable chance to be deceptive. We note that three of them were published alongside alternative classifiers that the respective authors considered preferable (5b, 6a, 14b). The probability that classifier 8 is deceptive is approximately 14% so we recommend to reevaluate it with a larger test set.

The split of the BM posterior into $r_{\text{informative}}$ and $r_{\text{deceptive}}$ in Eqs. (13) and (14) is a coarse graining device to ease conversation. A classifier with a very low absolute BM is neither informative nor deceptive but uninformative.

For finite $N$, $r_{\text{deceptive}}$ will be always greater than zero. What value of $r_{\text{deceptive}}$ can be tolerated will of course depend on the application scenario, and should be carefully considered by developers and users of classifiers.

## Large $N$, small difference in performance in meta-analysis of classifiers in machine learning

Our approach can also be used for meta-analyses of classifier ensembles, an application that is of considerable interest in machine learning (*Dietterich, 1998*; *Benavoli et al., 2017*; *Calvo et al., 2019*). Kaggle, a popular online community for machine learning challenges, provides a suitable environment for such meta-analyses. On Kaggle, participants build classifiers and submit their results online to be evaluated and compared to those of others. The best results are rewarded with cash prizes. The metric for evaluation depends on the individual challenge. Often, the competition is fierce and submitted results close, for example, accuracy sometimes differs by less than one per mille. With hundreds to tens of thousands of data points, test sets tend to be larger than in our literature collection above, but are still finite. Classifier metrics therefore retain some uncertainty, and statistical flukes could produce apparent differences in classifier performances that decide a competition.

We studied the Recursion Cellular Image Classification competition in greater detail (https://www.kaggle.com/c/recursion-cellular-image-classification/overview, accessed 31 January 2020 at 9:25 am CEST). Participants are tasked to properly classify biological signals in cellular images, disentangling them from experimental noise. Submissions were ranked based on multiclass accuracy. Micro-averaged multiclass accuracy can be modeled according to Eq. (5). We evaluated private leaderboards, that is, rankings provided by Kaggle with information on the participants and accuracies of their classifiers. These private leaderboards were also used to award prizes. Kaggle did not publish the exact size of the private test set but the overall test set contains 19,899 images and the private leaderboards were calculated on approximately 76% of it so we assumed $N = 15{,}123$. Based on $N$ and the published point estimates of ACC we could calculate TP + TN and FP + FN for every submitted classifier and compute a posterior distribution for ACC according to Eq. (5) (Fig. 5A).

These posterior distributions overlap. Using a Monte Carlo approach, we generated synthetic leaderboards from samples of the posterior distributions. Counting how often

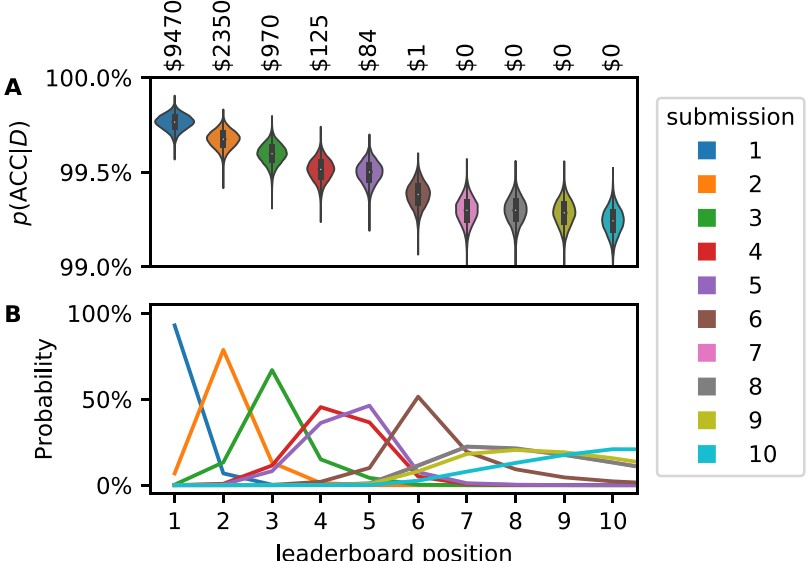

**Figure 5 Accuracy (ACC) posterior distribution for top ten submissions on Kaggle leaderboard (A). Distributions are narrow but the classifiers perform similarly. Therefore, after consideration of the uncertainty in ACC, the leaderboard positions of the submissions are uncertain (B).** If the cash prizes were awarded based on the probabilistic leaderboard, submissions outside of the top three would receive money (annotation). These estimates, too, are uncertain by a few percentage points.

every submission occurred at any leaderboard position yielded a probabilistic leaderboard (Fig. 5B). We observed that the winning submission has a 93% chance of being truly better than any other submission. For leaderboard position 4 and worse, rank uncertainty becomes considerable and ranking validity is limited by the sample size.

At the end of this competition, the top 3 submission were awarded $10.000, $2.000 and $1.000, respectively. This implies that it is certain that the submissions listed in the top 3 positions are indeed the best classifiers. As we have demonstrated, it is not certain which submissions are the best. If one would weigh the awarded prizes based on the probability of a submission to be in each rank, other participants would have been awarded small cash prizes (Fig. 5, top annotation).

Our approach is complementary to the Bayesian Plackett–Luce model, which considers multiple rankings for individual problems *Calvo et al. (2019)*. That model is agnostic about the performance metric since it is based only on the leaderboard position in every scenario. Consequently, it neglects the magnitude of the performance difference. Our approach on the other hand requires a generative model for the performance metric but works for individual problems and quantifies the performance gap between classifiers.

## Sample size determination

Since uncertainty in any commonly used metric decreases with increasing sample size $N$, we can employ our approach of uncertainty quantification also to determine in advance values of $N$ so that a classifier fulfills predefined MU criteria.

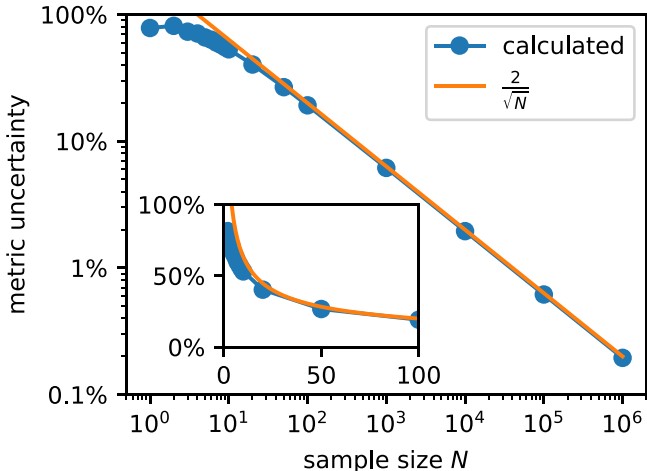

**Figure 6 Sample size determines metric uncertainty (defined by the length of the 95% highest posterior density interval) for any metric whose distribution follows a BBD.** Statistical power is 95%. The inset shows the same data on a non-logarithmic scale.

For those metrics which can be described as BBD (Eq. (5)), such as ACC, TPR, TNR and φ, we tested $N$ values spanning six orders of magnitude (Fig. 6), following Kruschke's protocol for sample size determination (*Kruschke, 2015b*). The shown results were obtained for a generating mode ω = 0.8 and concentration $k$ = 10. We found that different ω yielded almost indistinguishable results at low $k$.

The probability to achieve a MU more narrow than the given width in an empirical study, that is, statistical power, is 95%. The interpretation is as follows: If $N$ = 100, the likelihood that MU ≤ 19 percentage points is 95%. In order to decrease MU further, $N$ must be increased substantially.

Based on the standard deviation of a beta distribution and the central limit theorem we derive

$$\text{MU} \underset{\approx}{\leq} \frac{2}{\sqrt{N}} \tag{15}$$

for $N > 20$ in Section S5. It yields the correct order of magnitude which tells us if a classification study is feasible at the desired level of MU. This general rule ignores prior knowledge about the classifier. The posterior of the metric derived from exploratory classification experiments should be considered.

We found several papers presenting metrics with one or even two decimals. Classifier evaluations should be considered like any other experiment, and only significant digits should be given in their discussion. In Eq. (15) predicts that metric uncertainty would only drop below 0.1%, which is necessary to present a metric with a decimal, if the test data set included several million data points. Curating such a large test set is out of the question for the publications in our examples. On Kaggle leaderboards, ACC is presented as percentage with three decimals. Reducing metric uncertainty below 0.001% would require tens of billions of data points.

## CONCLUSIONS

In this work, we have presented a Bayesian model that quantifies the metric uncertainty of classifiers due to finite test sets. It is completely agnostic about the underlying classifier. Unlike previous work, our method cleanly separates data intrinsic $\phi$ from classifier intrinsic TPR and TNR, which facilitates transfer to different data sets. Nevertheless, our approach allows to evaluate metric uncertainty of all metrics that are based on the CM.

Our study of published examples suggests that MU is a neglected problem in classifier development. We found classifier metrics that were typically highly uncertain, often by tens of percentage points. The respective articles do not address this uncertainty, regularly presenting insignificant figures. Therefore, their audience is unintentionally mislead into believing that classifier metrics are known precisely although this is clearly not the case.

We could show that some classifiers carry a non-negligible risk of being deceptive. Moreover, empirical uncertainties, determined by repeating a classification experiment, would be even larger than the true uncertainty of a metric due to small $N$. Thus, many published classification metric point estimates are unlikely to be reproducible.

Poorly understood classifiers potentially harm individuals and society. Our example on cocaine purity analysis has shown that the number of miscarriages of justice due to an insufficiently tested classifier could be alarmingly high. Similarly, the likelihood of misdiagnoses and subsequent wrongfully administered therapies based on a medical classifier remain obscure unless we account for sample size. In basic science, uncertain classifiers can misguide further research and thus waste resources. During the identification of molecules with therapeutic potential, a poor classifier would discard the most promising ones or lead the researchers to a dead-end. Since time and funding are finite, this would decrease progress resulting in economic as well as medical damages.

The example of the Kaggle challenge shed light on the problem of uncertain performance in classifier meta-analysis. There, sample size is usually large but performance differences are minute. Consequently, classifier or algorithm rankings are uncertain.

We can interpret the frequent failure to account for metric uncertainty in classification as another facet of the current replication crisis, one root cause of which is neglect of uncertainty (*Gelman & Carlin, 2017*; *Wasserstein, Schirm & Lazar, 2019*). Classifier evaluation should be considered like any other experiment. It is obvious that a physical quantity cannot be measured exactly, and neither can a classifier metric. Thus, its uncertainty should be estimated and properly communicated.

For easy access to the method proposed here, we provide a free open-source software at https://github.com/niklastoe/classifier_metric_uncertainty. The software can be used without programing in an interactive web interface. The only required input is the confusion matrix, that is, information that is usually available for published classifiers. The software then computes the uncertainty for any of the commonly used classifier metrics. Moreover, sample sizes that are required to achieve a given exactness of a metric can be estimated according to Inequation 15. We hope this contributes to more realistic expectations, more thoughtful allocation of resources and ultimately reliable performance assessments of classifiers.

Our approach can be extended to similar problems. Multiclass classification can be modeled by $c + 1$ multinomial distributions (where $c$ is the number of classes), analogously to Fig. 1. Another extension of our approach is the computation of error bars of the popular receiver operating characteristic (ROC) curve, which is basically a vector of CMs. It would be more difficult to use our approach to compute the uncertainty of the area under the ROC curve (AUC), another popular classifier metric. However, the AUC, too, will be uncertain for finite $N$. A further extension is the inclusion of classification scores in a distributional model (*Brodersen et al., 2010b*), because the scores contain additional information that leads to a better understanding of MU.

Our approach only captures the uncertainty arising from finite $N$. Other sources of uncertainty such as over- or underfitting, data and publication bias etc. need to be considered separately. For instance, comparison of metric posterior distributions calculated separately for the training and test data could help to assess overfitting. Without such additional analyses, the posterior distributions obtained with our method are probably often too optimistic.

## ACKNOWLEDGEMENTS

We thank Paul Bürkner, Kai Horny, and Martin Theissen for fruitful discussion.

### Funding

This work was supported by the Deutsche Forschungsgemeinschaft through project CRC1093/A7. There was no additional external funding received for this study.
The funders had no role in study design, data collection and analysis, decision to publish, or preparation of the manuscript.

### Grant Disclosures

The following grant information was disclosed by the authors:
Deutsche Forschungsgemeinschaft: CRC1093/A7.

### Competing Interests

The authors declare that they have no competing interests.

### Author Contributions

- Niklas Tötsch conceived and designed the experiments, performed the experiments, analyzed the data, performed the computation work, prepared figures and/or tables, authored or reviewed drafts of the paper, and approved the final draft.
- Daniel Hoffmann conceived and designed the experiments, authored or reviewed drafts of the paper, and approved the final draft.

### Data Availability

Code is available at GitHub: https://github.com/niklastoe/classifier_metric_uncertainty.

We also provide a web-based tool so that users can determine metric uncertainties themselves without the need for programing experience at MyBinder: https://mybinder.org/v2/gh/niklastoe/classifier_metric_uncertainty/master?urlpath=%2Fvoila%2Frender%2Finteractive_notebook.ipynb.

## Supplemental Information

Supplemental information for this article can be found online at http://dx.doi.org/10.7717/peerj-cs.398#supplemental-information.

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
