# Peer review of "Classifier uncertainty: evidence, potential impact, and probabilistic treatment"

_PeerJ Computer Science, doi:10.7717/peerj-cs.398_

## Round 0.1 · original submission · Major Revisions

The article is acceptable for publication if the authors address the suggestions given by the reviewers (especially reviewer 2). Please, take it into account in the preparation of the new version of the manuscript.

Reviewer 1 ·

Basic reporting

The article was clearly written and professionally presented.
References were used properly.

Experimental design

Research questions were well defined, relevant, and meaningful.

Mathematical calculations were mentioned in a perfect way.

Validity of the findings

Conclusions were well stated and linked to the original research.

Additional comments

The manuscript was pretty impressive and they did a great job in explaining in a very beautiful manner.
I'm impressed and loved the content.

Reviewer 2 ·

Basic reporting

No comment

Experimental design

No comment

Validity of the findings

No comment

Additional comments

The authors proposed a metric for classifier uncertainty. I think the idea is important and novel. I have some suggestions to improve the study:
- Literature review are weak. The authors should add a substantial amount of related references to support their hypothesis and findings.
- The authors only tested their methods on a use case from Kaggle competition. It is not enough to convince the generality of the model. Thus I suggest the authors provide more use cases to make the work stronger.
- Evaluation metrics (i.e. accuracy, confusion matrix, ...) have been used in previous biological works with small dataset such as PMID: 33036150, PMID: 32942564, and PMID: 31987913. Therefore, the authors are suggested to refer more works to attract broader readership.
- The authors have not explained well on the classifiers that they used.
- Did the authors have some independent test on the results?

·

Basic reporting

No comment. The manuscript is well written and concise.

Experimental design

Normally, the sample size (24) and the wide spread of sizes (from 8 to 350) might be a concern, but the code being bundled allows for further verification.

Validity of the findings

The model defined for uncertainity quantification has been shown to arise from logical inconsistencies in the existing metrics (Caelen distributions). Furthermore, a full discussion of the prior considerations is also present. The fact that there exists large variation in the uncertainity of published classifier metrics is surprising, however, the analysis is valid and coherently presented.

---

## Round 0.2 · accepted · Accept

The reviewers and I agreed the paper is ready for publication. Congrats

Reviewer 2 ·

Basic reporting

no comment

Experimental design

no comment

Validity of the findings

no comment

Additional comments

no comment